# Factors Affecting Hospitalization and Death of Older Patients Who Need Long-Term Care—The Necessity of the Support for Dysphagia in Home Dental Care

**DOI:** 10.3390/geriatrics7020037

**Published:** 2022-03-25

**Authors:** Yoko Wakasugi, Chiaki Susa, Shino Murata, Jun Aida, Jun Sasaki, Junichi Furuya, Haruka Tohara

**Affiliations:** 1Department of Dentistry, Yushoukai Home Clinic, Yushoukai Medical Corporation, 5-14-10, Shimbashi, Minato, Tokyo 105-0004, Japan; yoko.venga.venga@gmail.com (Y.W.); chiaki.susa@gmail.com (C.S.); mrtshino@gmail.com (S.M.); 2Dysphagia Rehabilitation, Department of Gerontology and Gerodontology, Graduate School of Medical and Dental Sciences, Tokyo Medical and Dental University, 1-5-45, Yushima, Bunkyo, Tokyo 113-8549, Japan; furuyajunichi@gmail.com; 3Department of Oral Health Promotion, Graduate School of Medical and Dental Sciences, Tokyo Medical and Dental University, Tokyo 113-8549, Japan; aida.ohp@tmd.ac.jp; 4Yushoukai Home Clinic Headquarters, Yushoukai Medical Corporation, 5-14-10, Shimbashi, Minato, Tokyo 105-0004, Japan; junsasakimd@gmail.com; 5Department of Geriatric Dentistry, Showa University School of Dentistry, 2-1-1, Kitasenzoku, Ota, Tokyo 145-8515, Japan

**Keywords:** deglutition disorders, dental care, gerontology, home care services, long-term care

## Abstract

The demand for home dental care is increasing, but how it should be involved in the continuation of life at home for elderly people who need care has not been examined. Therefore, we examined whether items examined by dentists can affect hospitalization and death. The study included 239 patients with oral intake. They were divided into regular and non-regular diet groups, and ages, nutritional statuses, activities of daily living (ADLs), Charlson Comorbidity Indexes (CCI) and swallowing functions were compared. The nutritional statuses and ADLs of the three groups at the first visit and after one year were compared. The groups included those with stable, declined and improved diet forms. Factors influencing hospitalization and death over three years were examined. Nutritional status, swallowing function, CCI and ADLs were worse in the non-regular diet group. The declined diet form group had lower ADL levels and nutritional statuses at the first visit. A proportional hazards analysis showed significant differences in the changes in diet form for the stable and declined groups related to hospitalization (hazard ratio (HR): 6.53) and death (HR: 3.76). Changes in diet form were thought to affect hospitalization and death, and it is worthwhile to assess swallowing function in home dental care.

## 1. Introduction

In Japan, which has a super-aging society, home medical and dental care are practiced [1]. The goal is to continue living at home. Older people who need care often suffer from several illnesses and take multiple medications [2]. In addition, their cognitive function and activities of daily living (ADL) are impaired. They also have a higher risk of hospitalization and sudden changes in health compared to healthy people. Once hospitalized, their cognitive function, ADL and swallowing function deteriorate [3], making it difficult for them to be discharged to the place they used to live. Therefore, it is important to support daily life at home.

In Japan, home dental care became possible with the establishment of a new fee for home-visit care in 1988. Dentists often perform videoendoscopy (VE) [4] and the number of VEs performed during home dental care has increased since 2005 [5]. Thus, a system to support older patients undergoing medical treatment has been established. Currently, home dental care provides not only emergency dental care but also regular visits to older patients who need evaluations of swallowing function, including provisions of oral care and dental treatment, with the goals of preventing pneumonia and continuing oral intake.

Aging and decline in cognitive function affect oral health [6]. Dentistry is involved in the patient’s diet by providing dental care and assessing dysphagia. There is a need to predict the risk of hospitalization and death because food intake status and oral care are related to preventing hospitalization and supporting the enjoyment of eating to the end. However, only 8.2% of older adults were reported to receive home dental care [7], and research on home dental care mostly concerns the necessity of care and is of a cross-sectional design. The hospitalization of older patients who need care is often triggered by the exacerbation of chronic diseases, which are often accompanied by dysphagia [8]. To prevent unexpected hospitalization as much as possible, risk assessment from a dental perspective for dysphagia at home is necessary. If risk assessment, from a dental perspective, were possible, dentists could probably contribute to the prevention of hospitalization of older people who need care.

The purpose of this study was to determine whether items examined by dentists can affect hospitalization and death. We first conducted a cross-sectional survey to obtain insights into patients receiving home dental care. Next, a three-year prospective study was conducted.

## 2. Materials and Methods

Of the 272 patients aged 65 years or older who were visited from our institution (Yushoukai Medical Corporation, Yushoukai Home Clinic, Department of Dentistry, Tokyo, Home Care Support Clinic) between 1 April 2017 and 31 March 2020, 239 patients with oral intake were included in the study. Oral intake was defined as orally taking in all required nutrition from the diet. Patients were either homebound or institutionalized. Patients in the terminal stages of cancer or end-of-life care and those who were tube-fed or intravenously fed were excluded.

First, we conducted a cross-sectional study using data from the first visit. Three pre-calibrated dentists collected the data. We assessed age, sex, Charlson Comorbidity Index (CCI), Dysphagia Severity Scale (DSS) score [9] for swallowing function, Functional Oral Intake Scale (FOIS) score for food intake, Performance Status (PS) score for ADL and Mini Nutritional Assessment Short Form (MNA-SF) score [10] for nutritional status. In the MNA-SF, a score of 0–7 indicates malnutrition, 8–11 indicates a risk of malnutrition, and 12–14 indicates a good nutritional status. DSS was evaluated by conducting VE and being present at mealtime. MNA-SF was reevaluated once every 6 months, and DSS and FOIS were evaluated on a case-by-case basis when there were signs of dysphagia, but at least once every 6 months. Hospitalization and death during the 3-year follow-up were tracked, and we conducted a cohort study. We collected details about disease, hospitalization and death from the medical information supplied by physicians.

The oral intake group was divided into two groups: one eating a regular diet, with a FOIS score of 6–7 (the “regular diet group”), and the other eating a non-regular diet, with a FOIS score of 4–5 (the “non-regular diet group”). As a cross-sectional study of the two groups, we compared their ages, nutritional statuses, ADLs, and swallowing function. We also compared nutritional statuses and ADL levels at the time of initial diagnosis and one year later, dividing the patients into three groups: those whose FOIS scores remained the same (the “stable group”), those whose FOIS scores decreased by one or more levels (the “declined group”) and those whose FOIS scores improved (the “improved group”). As a prospective study, we examined the factors affecting hospitalization and death over three years.

In the cross-sectional studies, the χ-square test, Student’s *t*-test and Mann–Whitney U test were used for comparisons between the two groups (regular vs. non-regular diet groups), and the χ-square test and the Kruskal–Wallis test were used for comparisons among the three groups (stable vs. declined vs. improved groups). Proportional hazards analysis was used to examine factors affecting hospitalization and death. We wanted to determine whether age, nutritional status, ADL level, swallowing function, or change in diet form in a year had an effect on hospitalization and death (forced entry method). The sample size for the proportional hazards analysis was calculated for an observation period of three years, with survival rates of 0.38 for the hospitalized group and 0.78 for the control group, a power of 0.8 and a significance level of 0.05, which resulted in a sample of 54 cases.

This study was conducted according to the guidelines of the Declaration of Helsinki and approved by the Ethics Committee of the Yushoukai Medical Corporation (protocol code 003). Informed consent was obtained by means of an opt-out system on the website of the corporation. All statistical analyses were performed using SPSS Statistics (version 26.0; SPSS Inc., Chicago, IL, USA).

## 3. Results

### 3.1. Participants’ Characteristics

The study included 95 males and 144 females with a mean age of 84 ± 9 (69–104) years, the mean follow-up was 427 ± 302 (54–1160) days and the median follow-up was 347 days. The diseases recorded among the participants were dementia (37%), cerebrovascular disease (22%), degenerative disease (10%), chronic heart failure (9%), bone fracture (8%), chronic respiratory failure (4%), rheumatoid arthritis (2%) and others (8%).

### 3.2. Cross-Sectional Survey Results

Of the 239 patients, 163 (68.2%) were in the regular diet group and 76 (31.8%) were in the non-regular diet group. Their ages were not significantly different, but their MNA-SF, PS, CCI and DSS scores were significantly different (Figure 1). Compared with the regular diet group, the non-regular diet group had poorer nutritional status and lower ADL. In addition, the non-regular diet group had worse swallowing function and severer CCI.

Regarding the FOIS score change after one year in the regular diet group, 65.6% were in the stable group, 12.9% were in the declined group, 3.1% were in the improved group and 18.4% died or discontinued (Table 1). Of the three groups, the declined group had a lower nutritional status and ADL at the time of the first visit. In the non-regular diet group, 38.2% were in the stable group, 6.6% were in the declined group and had decreased diet form or were tube- or intravenously fed, 26.3% were in the improved group and 28.9% died or discontinued. There was a significant difference in nutritional status between the improved and declined groups after one year.

### 3.3. Prospective Study

During the 3-year observation period, 55 patients died (23.0%) and 53 patients (22.2%) experienced at least one hospitalization.

There were significant differences in PS score, FOIS score change and DSS score comparisons based on hospitalization status (Table 2). The hospitalization group had lower PS, a higher frequency of FOIS score decline and lower DSS. Proportional hazards analysis showed a significant difference in FOIS change between the stable and declined groups (hazard ratio (HR): 6.53, 95% confidence interval (CI): 3.06–13.2) (Figure 2), suggesting that the risk of subsequent hospitalization was higher in patients with decreased diet form.

There were significant differences in MNA-SF and PS scores, age, FOIS score change, DSS score and CCI compared to the death group (Table 3). MNA-SF, PS, CCI and DSS scores were worse, and the frequency of FOIS score decline was higher in the death group. Proportional hazards analysis showed a significant difference in FOIS score change between the stable and declined groups (HR: 3.76, 95% CI: 1.55–9.13) (Figure 2). The results suggest that a decrease in diet form is associated with mortality risk.

## 4. Discussion

### 4.1. Comparison of Diet Form

As a result of a cross-sectional survey to obtain insights into patients receiving home dental care, nutritional status, ADL and swallowing function were found to be lower in the non-regular diet group than in the regular diet group. These results are similar to those of previous studies, and a decreased diet form was associated with worse nutritional status [11,12]. In addition, diet form was affected by physical function as well as swallowing function [13,14]. CCI was also reported to be associated with severity of dysphagia [15]. In the regular diet group, within normal limits (DSS 7) and oral problems (DSS 5) were common, whereas in the non-regular diet group, oral problems (DSS 5) and fluid aspiration (DSS 3) were common. These results indicate that the need for nutritional and swallowing interventions was higher in the non-regular diet group.

In the regular diet group, one in every five patients experienced a decrease in diet form within the first year of dental intervention. The most common reasons for this were, in descending order, progression of Parkinson’s disease-related conditions, oral dysfunction, hospitalization-induced swallowing dysfunction, difficulty using dentures and pharyngeal stage disorders.

The nutritional status and ADL of the declined group were low at the first visit. The general condition of the patients was not uniform, even among those who ate regular meals, suggesting that nutritional intervention is necessary for patients who show a decline in ADL and nutritional status. This necessitates early assessment of the appropriateness of the diet, while subsequent changes in diet form may be predicted by assessing nutritional status and ADL.

In the non-regular diet group, 26% of patients improved their diet form within one year of the intervention. Considering that the nutritional status of the regular diet group was better than that of the non-regular diet group and that the improved and declined groups showed significant differences in nutritional status after one year, the results indicate that improvement in diet form is meaningful. It is important to maintain a functional oral cavity and change to a diet that matches oral function. There have been reports of changes due to swallowing interventions, such as a decrease in diet form following functional decline, and a report that patients who had been non-eating could regain the enjoyment of oral intake [5]. However, no reports have examined the frequency of diet form improvement due to dental interventions. The importance of dentures in swallowing function has been shown [16,17,18], and it is natural that dental treatment should improve diet form. This is an area which has seen remarkable improvements in home care patients. A 2002 Health Labor and Welfare Science study reported that 89.8% of the older population needed dental treatment or specialized oral care but only 26.9% received treatment. This study suggests that dental intervention could improve diet form. In addition, it is necessary to provide dietary support based on functional assessments, which should not end with dental treatment.

### 4.2. Factors Influencing Hospitalization and Death

Factors influencing the hospitalization of older people who need care were examined from a dental perspective and the results indicated that FOIS score change had an impact.

The hospitalization of older people who need care was associated with age, previous hospitalization, gender, comorbidity, living conditions, and behavioral factors (lack of exercise, falls, poor nutrition) [19]. In a study on the involvement of dysphagia, patients were divided into two groups based on DSS: those with and those without aspiration. There were 21.3% of patients in the aspiration group (DSS 1–4) and 78.7% of patients in the no-aspiration group (DSS 5–7). In a 4-year prospective study, hospitalization was significantly higher in the group with aspiration; DSS was associated with hospitalization, even after adjusting for age, sex, comorbidity index, Barthel Index and MNA-SF [8]. In our study, decreased diet form was listed as a significant factor, and decreased diet forms within 1 year were associated with a 6.35-fold increased risk of hospitalization. Although there are various reasons for hospitalization, diet is likely to affect hospitalization because it is related to nutritional status and ADL. In order to prevent hospitalization, it is necessary not to overlook changes in swallowing function and decline in oral function and share information on decline in diet form as a high risk for hospitalization.

In addition, it is necessary to assess DSS score at the time of first intervention so that interventions for dysphagia can be implemented before the primary complaint of dysphagia arises. In this study, 21.8% of the patients had a DSS score of 1–4, which was slightly higher than previously reported figures of 21.3% [8] and 9.4% [20]. Of the patients in this study, 54.8% had a DSS score of 5 or lower. Early assessment is desirable to avoid overlooking room for improvement or signs of deterioration.

Nutritional status affects the life prognosis of older people who need care [21,22,23]. In this study, even after adjusting for age and MNA-SF, we found that decreased diet form could be a predictor of death, and it was thought that decreased diet form reflects systemic changes.

Interventions during home dental care are aimed at maintaining food intake and preventing pneumonia and hospitalization. In addition to dental treatment, the need for functional assessments to provide dietary support was demonstrated. Malnutrition is common in older adults who need long-term care. One-fourth of the patients who are nutritionally at risk do not receive nutritional support, despite having been in contact with health care professionals [12]. We suggest, therefore, that dentists providing home dental care attend not only to dental care but look at the function of eating as well. As one possibility, it would be desirable for dentists to work with speech–language pathologists and nurses. 

The limitations of this study are that it only followed up changes in FOIS change one year after intervention and that no more detailed assessment of general condition than CCI was made.

## 5. Conclusions

For patients receiving home dental care, we aimed to determine whether health factors examined by dentists can affect hospitalization and death. More than half of the patients included in the study had a DSS of 5 or less at the time of the first visit, and both swallowing function and nutritional status were poor in the non-regular diet group. Given that some patients can improve their diet form and nutritional status through dental interventions, it is important to preserve teeth and make functional dentures to prevent deterioration in diet form. It is also important to provide dietary support based on functional assessment. Since changes in diet forms could have an impact on hospitalization and death, it is meaningful to evaluate swallowing function in addition to dental care in the provision of home dental care.

## Figures and Tables

**Figure 1 geriatrics-07-00037-f001:**
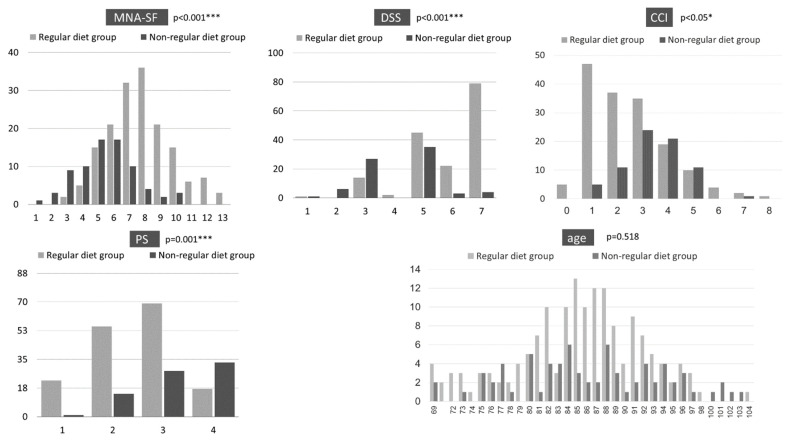
Cross-sectional survey results. * *p* < 0.05, *** *p* < 0.001. MNA-SF: Mini Nutritional Assessment Short Form; DSS: Dysphagia Severity Scale; PS: Performance Status; CCI: Charlson Comorbidity Index.

**Figure 2 geriatrics-07-00037-f002:**
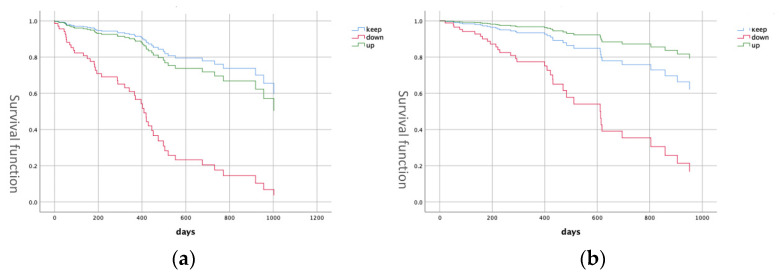
Survival curve. (**a**) Survival curve for hospitalization associated with FOIS change. (**b**) Survival curve for death associated with FOIS change. FOIS: Functional Oral Intake Scale.

**Table 1 geriatrics-07-00037-t001:** ADL and nutritional status of the first visit and one year later.

Regular Diet Group					Non-Regular Diet Group			
	stable	declined	improved		stable	declined	improved	
	*n* = 107	*n* = 21	*n* = 5		*n* = 107	*n* = 21	*n* = 5	
	(65.6%)	(12.9%)	(3.1%)		(65.6%)	(12.9%)	(3.1%)	
**PS**				*p* < 0.001 ***				*p* = 0.186
1	16 (15%)	0	1 (20%)	improved-declined	0	1 (20%)	0	
2	40 (37%)	3 (14%)	3 (60%)	*p* = 0.01 **	5 (17%)	0	6 (30%)	
3	46 (43%)	10 (48%)	0	stable-declined	11 (38%)	1 (20%)	10 (50%)	
4	5 (5%)	8 (38%)	1 (20%)	*p* < 0.001 ***	13 (45%)	3 (60%)	4 (20%)	
**MNA-SF**								*p* = 0.239
Malnutrition	40 (37%)	14 (67%)	2 (40%)	*p* = 0.007 **	23 (79%)	5 (100%)	18 (90%)	
At risk	59 (55%)	7 (33%)	3 (60%)	stable-declined	6 (21%)	0	2 (10%)	
Good	8 (7%)	0	0	*p* = 0.002 **	0	0	0	
**PS-1y**				*p* < 0.001 ***				*p* = 0.101
1	14 (13%)	0	1 (20%)	improved-declined	0	0	0	
2	40 (37%)	1 (5%)	3 (60%)	*p* = 0.04 *	5 (17%)	0	8 (40%)	
3	46 (43%)	11 (52%)	0	stable-declined	13 (45%)	2 (40%)	8 (40%)	
4	7 (7%)	9 (43%)	1 (20%)	*p* < 0.001 ***	11 (38%)	3 (60%)	4 (20%)	
**MNA-SF-1y**				*p* < 0.001 ***				
Malnutrition	45 (42%)	16 (76%)	1 (20%)	improved-declined	23 (79%)	5 (100%)	13 (65%)	*p* = 0.031 *
At risk	55 (51%)	5 (24%)	4 (80%)	*p* = 0.04 *	6 (21%)	0	6 (30%)	improved-declined
Good	7 (7%)	0	0	stable-declined	0	0	1 (5%)	*p* = 0.02 *
				*p* < 0.001 ***				

* *p* < 0.05, ** *p* < 0.01, *** *p* < 0.001. PS: Performance Status; MNA-SF: Mini Nutritional Assessment Short Form; PS-1y: PS of one year later; MNA-1y: MNA of one year later.

**Table 2 geriatrics-07-00037-t002:** Factors affecting hospitalization.

	No Hospitalization *n* = 186 (77.8%)	With Hospitalization *n* = 53 (22.2%)	Univariate	Proportional Hazard		
				*p* Value	Hazard Ratio	95% CI
MNA-SF	1:1 2:3 3:9 4:12 5:22 6:27 7:31 8:32 9:20 10:16 11:3 12:7 13:3	2:1 3:2 4:3 5:10 6:11 7:10 8:8 9:3 10:2 11:3	*p* = 0.111	*p* = 0.60	0.95	0.82–1.13
PS	1:20 (10.8%) 2:58 (31.2%) 3:76 (40.9%) 4:32 (17.2%)	1:3 (5.7%) 2:11 (20.8%) 3:21 (39.6%) 4:18 (34.0%)	* p * = 0.007 **	* p * = 0.17	1.43	0.86–2.37
Age	84.8	83.2	* p * = 0.272	* p * = 0.89	0.99	0.97–1.03
FOIS Change	stable: 114 (80.3%) declined: 8 (5.6%) improved: 20 (14.1%)	stable: 22 (48.9%) declined: 18 (40.0%) improved: 5 (11.1%)	* p * < 0.001 ***	*p* < 0.001 *** stable-improved: *p* = 0.50 stable-declined: *p* < 0.001***	stable-improved: 0.71 stable-declined: 4.63	stable-improved: 0.26–1.93 stable-declined: 1.65–13.0
DSS	1:1 (0.5%) 2:3 (1.6%) 3:28 (15.1%) 4:1 (0.5%) 5:63 (33.9%) 6:22 (11.8%) 7:68 (36.6%)	1:1 (1.9%) 2:3 (1.6%) 3:13 (15.1%) 4:1 (1.9%) 5:17 (32.1%) 6:3 (5.7%) 7:15 (28.3%)	* p * = 0.024 *	* p * = 0.12	0.83	0.65–1.05
CCI	0:5 (2.7%) 1:46 (24.7%) 2:35 (18.8%) 3:45 (24.2%) 4:31 (16.7%) 5:18 (9.7%) 6:4 (2.2%) 7:1 (0.5%) 8:1 (0.5%)	0:0 1:7 (13.2%) 2:15 (28.3%) 3:16 (30.2%) 4:10 (18.9%) 5:3 (5.7%) 6:0 7:2 (3.7%) 8:0	* p * = 0.188	* p * = 0.42	0.89	0.68–1.05

* *p* < 0.05, ** *p* < 0.01, *** *p* < 0.001. MNA-SF: Mini Nutritional Assessment Short Form; PS: Performance Status; FOIS: Functional Oral Intake Scale; DSS: Dysphagia Severity Scale; CCI: Charlson Comorbidity Index.

**Table 3 geriatrics-07-00037-t003:** Factors affecting death.

	Survival *n* = 184 (77.0%)	Death *n* = 55 (23.0%)	Univariate	Proportional Hazard		
				* p * Value	Hazard Ratio	95% CI
MNA-SF	2:2 3:5 4:8 5:23 6:26 7:34 8:36 9:20 10:15 11:6 12:7 13:2	1:1 2:1 3:5 4:7 5:9 6:12 7:8 8:5 9: 10:3 13:1	*p* < 0.001 ***	*p* = 0.20	0.88	0.72–1.07
PS	1:20 (11.0%) 2:59 (32.6%) 3:74 (40.9%) 4:28 (15.5%)	1:3 (5.5%) 2:10 (18.2%) 3:23 (41.8%) 4:19 (34.5%)	*p* = 0.001 **	*p* = 0.37	1.29	0.74–2.28
Age	83.6	87.5	*p* = 0.013 *	*p* = 0.23	1.03	0.98–1.07
FOIS Change	stable:115 (61.9%) declined: 16 (8.8%) improved: 23 (12.7%)	stable: 21 (63.6%) declined: 10 (30.3%) improved: 2 (6.1%)	*p* = 0.007 **	*p* = 0.005 ** stable-improved: *p* = 0.34 stable-declined: *p* = 0.011 *	stable-improved: 2.05 stable-declined: 7.77	stable-improved: 0.47–8.91 stable-declined: 1.61–37.5
DSS	1:2 (1.1%) 2:4 (2.2%) 3:22 (11.6%) 5:61 (33.7%) 6:20 (11.0%) 7:75 (40.3%)	2:2 (3.6%) 3:17 (30.9%) 4:2 (3.6%) 5:19 (34.5%) 6:5 (9.1%) 7:10 (18.2%)	*p* < 0.001 ***	*p* = 0.19	0.83	0.64–1.09
CCI	0:5 (2.7%) 1:41 (22.2%) 2:44 (23.9%) 3:45 (24.5%) 4:32 (17.4%) 5:14 (7.6%) 6:3 (1.6%) 7:0 8:0	0:0 1:12 (21.8%) 2:7 (12.7%) 3:16 (29.1%) 4:9 (16.3%) 5:6 (10.9%) 6:1 (1.8%) 7:3 (5.5%) 8:1 (1.8%)	*p* = 0.026 *	*p* = 0.75	0.95	0.70–1.29

* *p* < 0.05, ** *p* < 0.01, *** *p* < 0.001. MNA-SF: Mini Nutritional Assessment Short Form; PS: Performance Status; FOIS: Functional Oral Intake Scale; DSS: Dysphagia Severity Scale; CCI: Charlson Comorbidity Index.

## Data Availability

Not available. Data is not open due to ethical problem.

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
