# Peer review of "Factors Affecting Hospitalization and Death of Older Patients Who Need Long-Term Care—The Necessity of the Support for Dysphagia in Home Dental Care"

_geriatrics, 2022, doi:10.3390/geriatrics7020037_

Round 1
Reviewer 1 Report
The authors investigated whether a dental perspective could predict patient survival and hospitalization after 3 years. I understand the significance of this research. Although there are some places in the manuscript that describe the role of dentistry, the data used in this study can be obtained without dentistry. This paper should be revised to be useful for medical staff involved in geriatrics. The following points should be corrected.
Introduction
There is no need to report in such detail about the current state of dental care in Japan. It should be stated only in relation to this study. On the contrary, the explanation of the purpose of this study is insufficient. The need for dentistry to predict the risk of patient hospitalization and death should be mentioned.
Consideration
Line 230-233: “In this study, even after adjusting for age and MNA-SF, we found that decreased diet form could be a predictor of death, indicating that decreased diet form reflects systemic changes.”
Since there are few investigation items in this study, it is unclear what the changes in diet form reflect. If you worked with other medical occupations, you should have been able to find out the best method with a lot of information on oral health, severity of chronic diseases, cognitive and physical function, medication status, etc.
Line 233-234: “We believe that it makes sense to consider a decline in diet form as an optimal indicator for implementing advance care planning.”
Changes in diet form after one year are unknown at the stage of a care planning, so it does not seem to be suitable as an optimal indicator for implementing advance care planning.
Reviewer 2 Report
A review of Geriatrics
This study addresses the factors that could affect death and hospitalization among older adults. While the topic is interesting and important, the methods and presentation of results need great improvements. Moreover, many points should be addressed by the authors.
Introduction:
The introduction is not supported with enough references. Please add references, especially to the first two paragraphs.
Second paragraph:
It could be useful to add a sentence highlighting the impact of ageing and decline in cognitive function on oral health as well.
The first sentence of the fourth paragraph:
Please elaborate on the findings of this study (reference 3) and how this is relevant to the context of this study.
Please provide a clear aim of the study at the end of the introduction.
Methods:
L74: Please give more brief details of the institution (name, city, general hospital or academic).
L77-78: “Patients were either homebound or institutionalized, and in the terminal stages of cancer or end-of-life care and those who were tube-fed or intravenously fed were excluded. “this sentence is not clear. It seems like inclusion and exclusion criteria are mixed up. Rephrase, please.
L79-85: Please add a brief explanation of the different tools used in the study and how they were measured.
The method needs to add more information on how the participants were followed up and what variables were measured again during follow-up.
The authors need to explicitly state how they tracked Hospitalization and death.
The presentation of the method is confusing. It seems there are two follow-up periods. One year follow-up for the Change in FOIS and three-year follow-up for the hospitalization and death. The authors need to rephrase divide the text to explicitly explain different timelines of the data collection.
The authors also need to add more information on the examination setting and who collected the data and the number of examiners.
I would also change “keep” “up” “down” groups to “stable” “improved” “declined”, respectively.
Results:
L112: I would change “patients” to “participants”.
L115: the methods did not include anything related to the collection of disease. Please add this to the methods and how this was collected.
L117: I would argue that “old age” is not a disease. Reconsider, please.
It would be useful to add a table that contains the participant characteristics stratified by the oral intake groups.
Tables and figures are of poor quality.
This is not the proper way for presenting Proportional hazards analysis results. The tables should be rearranged and should include columns for hazards ratios and confidence intervals for each variable. The reference group for each variable should be highlighted as well.
Please refer to this paper for guidance:
Bradburn, M.J., Clark, T.G., Love, S.B. and Altman, D.G., 2003. Survival analysis part II: multivariate data analysis–an introduction to concepts and methods. British journal of cancer, 89(3), pp.431-436
Discussion:
Start the discussion with a recap of the aim and the main findings of the study.
The authors should also address the limitation of the study.
Reviewer 3 Report
This is an excellent research journal article.
The authors have conducted novel and innovative research that has produced significant results and important conclusions on pages 6-8.
It is also nicely written with clear and concise language
Suggestions:
Page 6, Line 150: Move Figure 2-1 and 2-2 to the left so that they fit on the page properly.
Page 8, Line 242: Write Speech Language Pathologists in full as these professionals are called differently in different countries.
Round 2
Reviewer 2 Report
I would like to thank the authors for incorporating my comments within the revised version of the manuscript. the manuscript has significantly improved. However, two minor points should be addressed.
1) L56: I would change "figure out" to "assess" or "examine".
2)L87: add"data" after "we collected".
This manuscript is a resubmission of an earlier submission. The following is a list of the peer review reports and author responses from that submission.
Round 1
Reviewer 1 Report
This manuscript examines which factors are likely to affect hospitalization and death among those who require long-term care “from the perspective of home dental care.” However, the dental care aspect in this manuscript is lost, or ill-defined despite the title of the paper. This reader was expecting to see an examination of common dental procedures/care provided and hospitalizations and/or death, however, it seems that the comparison has little to do with dental care. The authors mention that VE (video endoscopy) is commonly performed by dentists in this population in Japan, however, VE is not usually considered a “dental” procedure. The authors should reconsider their title given their methods and reported findings.
In general, the manuscript would greatly benefit from some editing for clarity and grammar. Some sentences do not follow a logical order, and some explanations are completely missing (see below for examples).
For clarity, perhaps the first paragraph on the second page should be reworded to be more clear:
“In Japan, home dental care became possible with the establishment of a new fee for 45 home-visit care in 1988. In Japan, dentists often perform videoendoscopy (VE)[2] and the number of VE performed during home dental care has increased since 2005 [1]. Thus, a system to support older patients undergoing medical treatment has been established.”
The authors should define “oral intake” in the second sentence of the Materials and Methods section. In the second sentence of the Results, section, the authors describe the “main 108 complaints about home dental care” however this question was not previously described in the Materials and Methods – was this a questionnaire administered to the home-bound elders or to their caregivers? The authors should describe this questionnaire in the Materials and Methods Section.
What is the “intervention” that shows up first in Figure 1? There is no description of this intervention and what this part of the figure shows. In Tables 2 and 3, the authors should define how one should interpret the MNA as readers may not be familiar with these notations. In Tables 2 and 3, the keys should define all acronyms.
The findings and conclusions are not surprising and seem to reflect what has been shown in previous studies; it is unclear what this manuscript is adding to the knowledge base. If there is something novel, the authors should make it clear what their study adds.
Author Response
We thank reviewers for fruitful suggestions, especially for suggesting better terms and sentences.
Response to Reviewer 1:
We wish to express our appreciation to the reviewer for the insightful comments, which have helped us significantly improve the paper.
Comment 1:
This manuscript examines which factors are likely to affect hospitalization and death among those who require long-term care “from the perspective of home dental care.” However, the dental care aspect in this manuscript is lost, or ill-defined despite the title of the paper. This reader was expecting to see an examination of common dental procedures/care provided and hospitalizations and/or death, however, it seems that the comparison has little to do with dental care. The authors mention that VE (video endoscopy) is commonly performed by dentists in this population in Japan, however, VE is not usually considered a “dental” procedure. The authors should reconsider their title given their methods and reported findings.
Response: We thank the referee for this pertinent comment. Japan is super aged society, so dentists also visit the patients’ homes and treat for the enjoyment of eating and assess their oral function. Unlike outpatient clinics, home dental care is not complete with dental treatment alone, but requires that the patient is able to eat and maintain nutritional status. Therefore, it is necessary to evaluate the general condition of the patient. It is obvious that the oral condition of the patient will improve as a result of the dental intervention, but we would like to show what home dental care can do to actually support their lives. We are conducting clinical research because we want people to know about such home dental care in Japan, so we revised the title.
Comment 2:
In general, the manuscript would greatly benefit from some editing for clarity and grammar. Some sentences do not follow a logical order, and some explanations are completely missing (see below for examples).
For clarity, perhaps the first paragraph on the second page should be reworded to be more clear:
“In Japan, home dental care became possible with the establishment of a new fee for home-visit care in 1988. In Japan, dentists often perform videoendoscopy (VE)[2] and the number of VE performed during home dental care has increased since 2005 [1]. Thus, a system to support older patients undergoing medical treatment has been established.”
Response: We thank the referee for this pertinent comment. We revised the paper and we sent it out for proofreading again.
Comment 3:
The authors should define “oral intake” in the second sentence of the Materials and Methods section. In the second sentence of the Results, section, the authors describe the “main complaints about home dental care” however this question was not previously described in the Materials and Methods – was this a questionnaire administered to the home-bound elders or to their caregivers? The authors should describe this questionnaire in the Materials and Methods Section.
Response: We thank the referee for this pertinent comment. We added the definition of oral intake in the Materials and Methods section. We deleted the main complaints.
Comment 4:
What is the “intervention” that shows up first in Figure 1? There is no description of this intervention and what this part of the figure shows.
In Tables 2 and 3, the authors should define how one should interpret the MNA as readers may not be familiar with these notations.
In Tables 2 and 3, the keys should define all acronyms.
Response: We thank the referee for this pertinent comment. This study is not an intervention study, so it is not listed. I found the presence of swallowing intervention confusing, so I deleted it.
We added the MNA evaluation in the Materials and Methods section.
In Tables 2 and 3, We revised.
Comment 5:
The findings and conclusions are not surprising and seem to reflect what has been shown in previous studies; it is unclear what this manuscript is adding to the knowledge base. If there is something novel, the authors should make it clear what their study adds.
Response: We thank the referee for this pertinent comment. Few studies have prospectively followed patients in home dental care, and we believe that our study is new because it provides suggestions for practicing dentistry in home dental care while assessing the risk of hospitalization and prognosis. Moreover, it is a novelty that home dental care including the support for swallowing disorders is beneficial.
Reviewer 2 Report
It is difficult to decipher the main research question of this study. If the question is "Home dental assessments decrease hospitalizations and death" the paper did not have any data related to answering this question. More specifically there was no control group that did not get dental assessments.
There was a lot of descriptive data about the patient assessment but no connection between the descriptive data and the intervention (dental assessments). Without this connection I do not believe this paper can be saved.
Other problems in the paper include generalizations that are not substantiated by citations for example in lines 35-38.
Lines 39-44 include sweeping generalizations such as that all older people who need care are cognitively and functionally impaired.
Line 82 describes a group eating non-regular food without defining what this means. Please define this group. Were they eating mechanical soft? puree? vegetarian? or what?
Defining this group is important because in Line 118-120 they are noted to have all these differences such as poor nutritional status and worse swallowing functions but if the patient on "non regular" diet is puree to start with then it is obvious the patient would have swallow dysfunction. If the "non regular" diet is low calorie to start with then it is obvious the patient would have poorer nutritional status.
Author Response
We thank reviewers for fruitful suggestions, especially for suggesting better terms and sentences.
Response to Reviewer 2:
We wish to express our appreciation to the reviewer for the insightful comments, which have helped us significantly improve the paper.
Comment 1:
It is difficult to decipher the main research question of this study. If the question is "Home dental assessments decrease hospitalizations and death" the paper did not have any data related to answering this question. More specifically there was no control group that did not get dental assessments. There was a lot of descriptive data about the patient assessment but no connection between the descriptive data and the intervention (dental assessments). Without this connection I do not believe this paper can be saved.
Response: We thank the referee for this pertinent comment. CQ is what factors dentists can assess affect hospitalization and death in older patients who need long-term care. This study is not an intervention study. If this study was an intervention study, I should compare patients with or without dental intervention, but I didn’t intend that, so I did not compare these two groups.
Comment 2:
Other problems in the paper include generalizations that are not substantiated by citations for example in lines 35-38.
Response: We thank the referee for this pertinent comment. We revised.
Comment 3:
Lines 39-44 include sweeping generalizations such as that all older people who need care are cognitively and functionally impaired.
Response: We thank the referee for this pertinent comment. This is the Japanese system of insurance for long-term care. The older people who can use the insurance are cognitively and functionally impaired and they can use home dental care.
Comment 4:
Line 82 describes a group eating non-regular food without defining what this means. Please define this group. Were they eating mechanical soft? puree? vegetarian? or what?
Defining this group is important because in Line 118-120 they are noted to have all these differences such as poor nutritional status and worse swallowing functions but if the patient on "non regular" diet is puree to start with then it is obvious the patient would have swallow dysfunction. If the "non regular" diet is low calorie to start with then it is obvious the patient would have poorer nutritional status.
Response: We thank the referee for this pertinent comment. In Japan, the diet form for swallowing is defined that regular diet is rice and normal diet, and non-regular diet is rice gruel and chopped food or purée. In Japan, even if the patients eat non-regular diet at home, they sometimes don’t have swallow dysfunction because they are ordered to eat them during hospitalization. Even if the patients eat regular diet, they sometimes show malnutrition because they eat only carbohydrates or eat a little for a variety of reasons, such as loss of appetite and oral hypofunction.
Reviewer 3 Report
The authors examined whether one-year changes in diet can predict survival after three years. To this end, the authors perform Propotional hazards analysis using Performance Status (PS), MNA-SF, Age, DSS, and FOIS change as explanatory variables.
Since FOIS change shows the change for one year, it is an index that includes the influence of various changes (decrease in ADL, seriousness of disease, deterioration of cognitive function, etc.). On the other hand, PS, MNA-SF, and DSS are all measured values ​​at the time of the first visit, so it is not possible to adjust for these changes that occurred during one year. Therefore, it cannot be said that changes in diet form directly affected hospitalization and death of the subjects in this study. The abstract and discussions need to be revised to explain this to the reader.
In addition, the following items need to be revised.
Introduction
This study does not aim to verify the effects of dental interventions. Therefore, it is not necessary to elaborate on why dentistry began to perform VE. Instead, you should describe the findings that have been clarified so far regarding the prognosis of care recipients.
Materials & Methods
What was the procedure for changing the diet form of the subjects? What role does dentistry play in determining diet form?
Results
It is necessary to describe the MNA-SF, PS, DSS, Age, and disease of the keep group, down group, and up group, respectively. If there is a significant bias, you should consider the impact on this study results. There are 55 people in the death group, but 52 people died or discontinued within 1 year, and 33 people died within 1 to 3 years, does that mean that 30 subjects discontinued within 1 year? Details should be provided.
Figure 1
Please also graph age and PS.
Table 1
A mixture of those with and without asterisks.
A mixture of uppercase and lowercase letters.
MNA should be described as MNA-SF.
Table 2
A mixture of those with and without asterisks.
A mixture of uppercase and lowercase letters.
MNA should be described as MNA-SF.
The abbreviation is insufficient.
Table 3
A mixture of those with and without asterisks.
A mixture of uppercase and lowercase letters.
MNA should be described as MNA-SF.
The abbreviation is insufficient.
The Survival group is 184 patients, but the total number listed in MNA, PS, and DSS is 181 patients.
Figure 2
The font size of the graph is too small.
References
Please arrange the writing style of the author's name.
Author Response
We thank reviewers for fruitful suggestions, especially for suggesting better terms and sentences.
Response to Reviewer 3:
We wish to express our appreciation to the reviewer for the insightful comments, which have helped us significantly improve the paper.
Comment 1:
The authors examined whether one-year changes in diet can predict survival after three years. To this end, the authors perform Propotional hazards analysis using Performance Status (PS), MNA-SF, Age, DSS, and FOIS change as explanatory variables. Since FOIS change shows the change for one year, it is an index that includes the influence of various changes (decrease in ADL, seriousness of disease, deterioration of cognitive function, etc.). On the other hand, PS, MNA-SF, and DSS are all measured values ​​at the time of the first visit, so it is not possible to adjust for these changes that occurred during one year. Therefore, it cannot be said that changes in diet form directly affected hospitalization and death of the subjects in this study. The abstract and discussions need to be revised to explain this to the reader.
Response: We thank the referee for this pertinent comment. We thought the change in diet form may be associated with hospitalization and death, so we chose it. We want to predict the risk of hospitalization and death, and we think it will begin with a change in diet form. As you have pointed out, we revised.
Comment 2:
Introduction
This study does not aim to verify the effects of dental interventions. Therefore, it is not necessary to elaborate on why dentistry began to perform VE. Instead, you should describe the findings that have been clarified so far regarding the prognosis of care recipients.
Response: We thank the referee for this pertinent comment. The three patterns of death are those. The first is short period of evident decline—typical of cancer. Most patients with malignancies maintain comfort and functioning for a substantial period. However, once the illness becomes overwhelming, the patient’s status usually declines quite rapidly in the final weeks and days preceding death. The second is long-term limitations with intermittent exacerbations and sudden dying—typical of organ system failure. Patients in this category often live for a relatively long time and may have only minor limitations in everyday life. From time to time, some physiological stress overwhelms the body’s reserves and leads to a worsening of serious symptoms. Patients survive a few such episodes but then die from a complication or exacerbation, often rather suddenly. Ongoing disease management, advance care planning, and mobilizing services to the home are key to optimal care. The third is prolonged dwindling—typical of dementia, disabling stroke, and frailty. Those who escape cancer and organ system failure are likely to die at older ages of either neurological failure (such as Alzheimer’s or other dementia) or generalized frailty of multiple body systems. Supportive services at home, like Meals on Wheels and home health aides, then institutional long-term care facilities are central to good care for this trajectory.
Citation: J Lynn and DM Adamson. Living well at the end of life. Adapting health care to serious chronic illness in old age, CA, Rand corporation, 2003
Comment 3:
Materials & Methods
What was the procedure for changing the diet form of the subjects? What role does dentistry play in determining diet form?
Response: We thank the referee for this pertinent comment. First, we evaluate the general condition of the patient, including medical history, medications, and physical functions. Next, we assess the oral function, the time required for a meal, and the amount of dietary intake. In the case of disease which occurs pharynx stage disorder, we use videoendoscopy.
Comment 4:
Results
It is necessary to describe the MNA-SF, PS, DSS, Age, and disease of the keep group, down group, and up group, respectively. If there is a significant bias, you should consider the impact on this study results.
There are 55 people in the death group, but 52 people died or discontinued within 1 year, and 33 people died within 1 to 3 years, does that mean that 30 subjects discontinued within 1 year? Details should be provided.
Response: We thank the referee for this pertinent comment.
Regular diet group
|
up (N=5) |
keep (N=107) |
down (N=21) |
||
|
MNA-SF |
7:2,9:2, 10:1 |
3:1, 4:1, 5:8, 6:15, 7:15, 8:26, 9:16, 10:13, 11:4, 12:6, 13:2 |
3:1, 4:3, 5:3, 6:2, 7:5, 8:5,9:1,11:1 |
p=0.007 |
|
PS |
1:1, 2:3, 4:1 |
1:16, 2:40, 4:46, 4:5 |
2:3, 3:10, 4:8 |
p<0.001 |
|
DSS |
5:5 |
1:1,3:4,5:27,6:18,7:57 |
3:6,4:1,5:4,6:2,7:8 |
p=0.005 |
|
age |
78.2±6.4 |
83.2±9.2 |
82.2±8.6 |
P=0.45 |
Non-regular diet group
|
up (N=20) |
keep (N=29) |
down (N=5) |
||
|
MNA-SF |
3:2,4:3,5:3,6:7,7:3,8:1,10:1 |
2:1,3:3,4:2,5:7,6:5,7:5,8:3,9:1,10:2 |
2:1,4:1,5:2,6:1 |
P=0.24 |
|
PS |
2:6,3:10,4:4 |
2:5,3:11,4:13 |
1:1,3:1,4:3 |
P=0.016 |
|
DSS |
2:1,3:5,5:10,6:2,7:2 |
1:1,2:2,3:10,5:15,7:1 |
2:1,3:3,5:1 |
P=0.068 |
|
age |
85.2±8.9 |
85.5±9.0 |
81.4±11.3 |
p=0.63 |
PS was significantly different in both groups. The analysis shows that the impact is small, so it is not shown in the paper.
Thirty-six died in the first year, 15 in the second year, and 4 in the third year.
Comment 5:
Figure 1
Please also graph age and PS.
Response: We thank the referee for this pertinent comment. We revised.
Table 1
A mixture of those with and without asterisks.
A mixture of uppercase and lowercase letters.
MNA should be described as MNA-SF.
Response: We thank the referee for this pertinent comment. We revised.
Table 2
A mixture of those with and without asterisks.
A mixture of uppercase and lowercase letters.
MNA should be described as MNA-SF.
The abbreviation is insufficient.
Response: We thank the referee for this pertinent comment. We revised.
Table 3
A mixture of those with and without asterisks.
A mixture of uppercase and lowercase letters.
MNA should be described as MNA-SF.
The abbreviation is insufficient.
The Survival group is 184 patients, but the total number listed in MNA, PS, and DSS is 181 patients.
Response: We thank the referee for this pertinent comment. We revised.
Figure 2
The font size of the graph is too small.
Response: We thank the referee for this pertinent comment. We revised.
References
Please arrange the writing style of the author's name.
Response: We thank the referee for this pertinent comment. We revised.
Round 2
Reviewer 2 Report
The author does not provide an adequate response to the initial concern about what is the main research question in this study.
Reviewer 3 Report
The authors consistently state the importance of dental interventions in this paper. However, there is no evidence from this study to support its importance. The authors argue that it is important to check for changes in dietary patterns after one year, thanks to dental intervention. However, it is possible for non-dentists to record changes in dietary patterns. The reader cannot understand without stating why it should be a dental assessment.
It is interesting to be able to predict the likelihood of hospitalization or death after 3 years by assessing some functions of the patient. Based on the results, it should be discussed not about the importance of dental interventions, but about some of their functions. To do this, the structure of the text needs to be significantly modified.